

# Seroprevalence of five diarrhea-related pathogens in bovine herds of scattered households in Inner Mongolia, China between 2019 and 2022

Mengyuan Xie, Kejia Chen, PingPing Liu, Xiaodan Wang, Yexin Chen, Hewei Shang, Yanru Hao, Peiyuan Gao, Xiuling He and Xiaojing Xu

Inner Mongolia Agricultural University, Hohhot, China
Ministry of Agriculture and Rural Affairs, Key of Laboratory of Clinical Diagnosis and Treatment Technology in Animal Disease, Hohhot, Inner Mongolia, China

Corresponding authors
Xiaojing Xu,
xuxiaojinglaoshi@163.com
Xiuling He, xiulinghe2008@163.com

## ABSTRACT

Bovine diarrhea is a multi-factorial disease and remains one of the biggest health problems in animal husbandry. The endemic trends of the main pathogens responsible for bovine diarrhea in Inner Mongolia have not been analyzed systematically before. Therefore, the purpose of this study was to estimate the prevalence of bovine diarrhea pathogens found in the scattered households of Inner Mongolia in China. Additionally, we assessed for differences in the prevalence of infection based on age and region, as well as determined local prevalence rates and the rates of mixed infections. Using a two-stage random sampling strategy, 3,050 serum samples were collected from 72 bovine herds in 11 leagues and cities in Inner Mongolia, and the positive rates of BVDV, BRV, BCoV, K99, and *Mycobacterium paratuberculosis* (*M. paratuberculosis*) antibodies in the samples were detected by ELISA to determine the epidemic trends and epidemic differences of the five pathogens in Inner Mongolia. The positive rates of antibodies based on serum samples were: BVDV, 18.79% (95% CI [17.44–20.22]); BRV, 12.39% (95% CI [11.27–13.61]); BCoV, 12.82% (95% CI [11.68–14.05]); K99, 13.80% (95% CI [12.62–15.07]); and *M. paratuberculosis*, 10.79% (95% CI [9.74–11.94]). The prevalence rates of BRV, BCoV and K99 at 0–2 months were significantly different from that at 2–6 months, 6–18 months and adult cattle ($P < 0.05$). The prevalence of BVDV and *M. paratuberculosis* was the highest in adult cattle, which was significantly different from that in other age groups ($P < 0.05$). Furthermore, obvious regional epidemiological differences among the five diseases were observed. There was a mixed infection of BRV+BCoV in each age stage, the highest mixed infection being BVDV+BRV+K99 at 0–2 months of age. Our results showed that the cattle of scattered households in the Inner Mongolia of China were endemically infected with several important cattle pathogens. Most of the pathogens studied occurred between 0–2 months of age and were mixed infections, which greatly influences the health of the cattle and leads to economic loss. These findings are of practical significance for the future prevention and control of bovine diarrhea in the Inner Mongolia or other regions of China.

## INTRODUCTION

Bovine diarrhea is a well-known disease in the bovine industry that causes substantial economic losses worldwide due to high morbidity, mortality, growth retardation, and treatment costs, as well as serious long-term consequences including delayed first calving (*EFSA Panel on Animal Health and Welfare, 2012*).

Bovine diarrhea is mainly caused by infectious and non-infectious factors such as pathogens, management, nutrition, and immune status (*Maes et al., 2003*; *Stipp et al., 2009*). The infectious factors typically consist of viral pathogens (such as bovine rotavirus (BRV), bovine coronavirus (BCoV), and bovine viral diarrhoea virus (BVDV)), and bacterial pathogens (such as *Escherichia coli* K99 (*E. coli* K99), *Mycobacterium paratuberculosis* (*M. Paratuberculosis*), and Salmonella). These pathogens can cause diarrhea and lead to death or unsatisfactory growth of the infected animals (*Cho & Yoon, 2014*; *Gillhuber et al., 2014*; *Scharnböck et al., 2018*). Moreover, during the occurrence of bovine diarrhea disease, it is difficult to determine the exact role of each pathogen, because their effects are not always isolated, and it often presents similar symptoms clinically. Moreover, epidemiological studies have shown that in the case of mixed infections, the diversity of the degree of association between these pathogens is often fatal (*Dahmani et al., 2020*; *Hubálek, Rudolf & Nowotny, 2014*; *Martella, Decaro & Buonavoglia, 2015*).

In animal husbandry, the livelihoods of farmers and herdsmen are unique in many aspects. For example, farmers and herdsmen use vast pastures, natural grassland vegetation, and crops to raise bovine. They are usually located in high-altitude areas and are easily affected by the weather in China (*Chenais et al., 2021*). Therefore, maintaining the animals' health is vital to the livelihoods of the families that rely on animal husbandry (*Chenais et al., 2021*; *Perry & Grace, 2009*).

Inner Mongolia of China is located in the golden zone of bovine breeding at 40–45° north latitude (*Li, 2022*). It is considered an internationally recognized high quality dairy breeding belt along with areas such as Europe, South America, and New Zealand. Relying on rich grassland resources, clean soil and water, and natural grazing methods, Inner Mongolia vigorously promotes the high-quality development of cattle breeding industry (*Grace et al., 2017*). In Inner Mongolia, bovine trade with other Chinese provinces and neighboring countries is a frequent and important economic activity, yet the pattern of endemic diseases is largely unknown. Previously, bovine diarrhea diseases occurred in this area, causing huge losses to the bovine industry in this region (*Gao, 2022*). Determining the cause of diarrhea is crucial for the provision of accurate treatment and preventive measures (*Ryu, Shin & Choi, 2020*). However, only sporadic investigations and analyses have been conducted on the pathogens causing bovine diarrhea in Inner Mongolia, China (*Li et al., 2021*). Currently, there is no representative study on the epidemiological characteristics of the main pathogens causing bovine diarrhea among scattered households, and the regional epidemiological differences and prevalence of mixed infections remain unclear. It is necessary to determine the pathogenic factors and epidemiological characteristics of bovine diarrhea in Inner Mongolia.

This study assessed the seroprevalence in Inner Mongolia, China, with the aim of analyse the prevalence of pathogens that cause diarrhea in cattle according to age in scattered households in 11 leagues and cities, to provide data for prevention and control of bovine diarrhea in Inner Mongolia.

## MATERIALS & METHODS

### Research design and herd selection

Inner Mongolia has 12 prefecture-level administrative regions, including nine prefecture-level cities and three leagues, in the herd recruitment stage, the Alxa League was excluded because the Gobi desert area of the Alxa League is large with limited cattle breeding.

According to the visit and investigation of the scattered households in Inner Mongolia and the data provided by the Inner Mongolia Animal Disease Control Center, the breeding scale of scattered households is mostly 10–300 heads, so the herd and sample collection are carried out within this range. Herd and samples were randomly selected in two-stage. In the first stage, scattered households were selected from the whole area, in the second stage, a sampling strategy based on disease occurrence was adopted for individual sampling, and the parameters were set to 95% confidence level (CI) and 30% expected prevalence (P), with a statistical error set to 5%.

A total of 91 bovine herds were studied from 11 leagues and cities. Among them, 19 bovine herds were deleted from the sample group due to an absence of diarrhea or a lack of interest in participating in the study. With mutual consent, the bovine herds were visited once and recorded the herds living environment, feeding, management methods, vaccination, and clinical symptoms.

The relative herd size positions of 72 scattered households in 11 leagues and cities of Inner Mongolia were visited and collected samples between June 2019 to May 2022 (as shown in Fig. 1, the size of the dots and the color intensity correlated with the number of herds).

### Sample collection and clinical examination

Animals age, symptoms of diarrhea, and vaccination status were recorded under the condition of mutual agreement.

A 5 mL vacuum tube without additives and a disposable venous blood sampling needle were used to collect blood samples from the jugular vein of calves and the tail vein of adult cattle. The samples were refrigerated at 4 °C prior to being sent to the laboratory. The serum was then separated and frozen at −20 °C until serological testing. Table 1 summarizes the ELISA kits information.

### Detection of antibodies to diarrhea pathogens in blood samples

According to the kit instructions (Table 1), commercial enzyme-linked immunosorbent assay to detect whether there are BVDV, BRV, BCoV, *E. coli* K99, and *M. paratuberculosis* antibodies in serum samples. The OD value of samples was detected by the microplate reader marker at 450 nm, all enzyme-linked immunosorbent test results were classified as negative or positive according to ELISA kits instructions and test cut-off value.

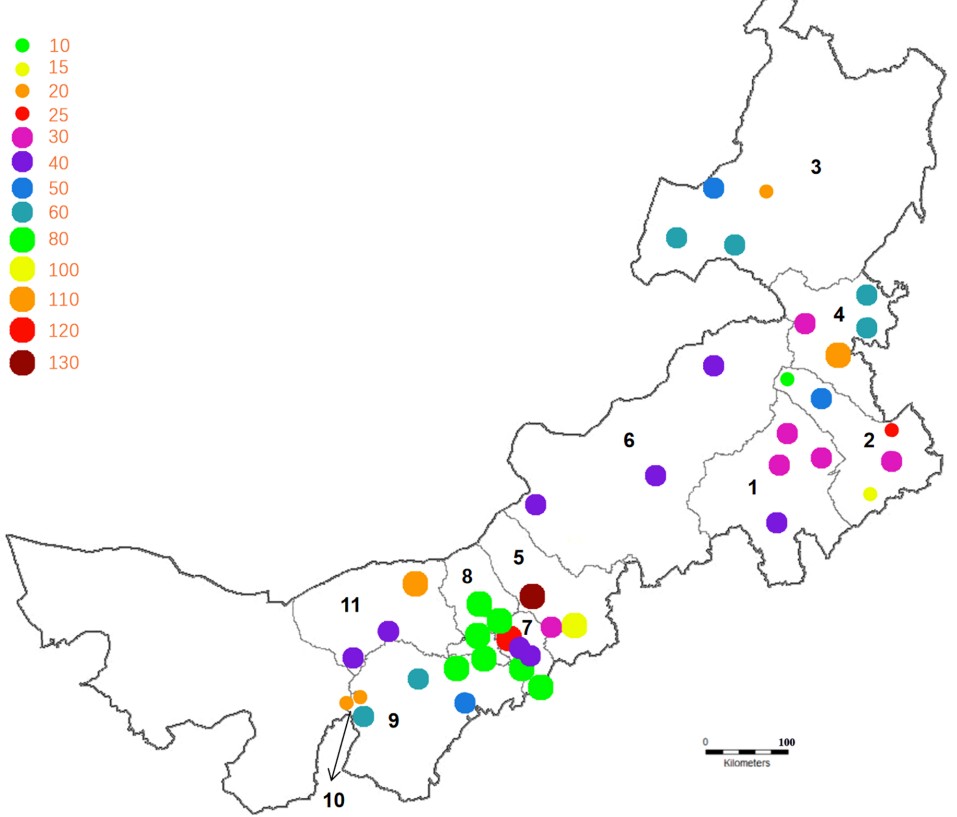

**Figure 1** **The distribution location and scale of the bovine herds.** Note: Different colors and sizes represent the scale of sampling in each league. The numbers 1–11 represent the 11 leagues and cities in Inner Mongolia: 1 is Chifeng, 2 is Tongliao, 3 is Hulun Buir, 4 is Hinggan League, 5 is Ulanqab, 6 is Xilingol League, 7 is Hohhot, 8 is Baotou, 9 is Ordos, 10 is Wuhai, 11 is Bayan Nur.

**Table 1** **Diagnostic test kits used for testing serum samples.**

| Pathogen | Diagnostic test kits |
| --- | --- |
| BVDV | BVDV Antibody Test Kit (IDEXX) |
| BRV | Bovine coronavirus ELISA kit (Bio-X Diagnosis) |
| BCoV | Bovine rotavirus ELISA kit (Bio-X Diagnosis) |
| *E.coli* K99 | *E.coli* F5 (K99) ELISA kit (Bio-X Diagnosis) |
| *M. paratuberculosis* | INgezim Paratuberculosis 12.PTB.K.1 (eurofins) |

ELISA technique:

1. All reagents must be allowed to come to 18–26 °C before use (except enzyme-labeled antibodies of BRV, BCoV and *E. coli* K99), Reagents should be mixed by gentle inverting or swirling;

2. Serum were multiply diluted (100-fold dilution for BCoV, 20-fold dilution for BRV and *M. paratuberculosis*, 2-fold dilution for *E. coli* K99, and 5-fold dilution for BVDV),

and the diluted sera for detection of *M. paratuberculosis* were preincubated for 30 min in sample dilution plate;

3. BVDV ELISA plates with 125 μL diluted serum per well, *M. paratuberculosis* and *E. coli* K99 ELISA plates with 100 μL of diluted serum per well, incubated at 18–26 °C (BVDV 90 min, *M. paratuberculosis* 45 min, *E. coli* K99 2 h), BCoV and BRV ELISA plates with 100 μL of diluted serum per well and then added 100 μL of enzyme-labeled antibody, add negative into duplicate wells, add positive into duplicate wells, incubated at 18–26 °C for 1 h;

4. Remove the solution and wash each well with approximately 300 μL of wash solution 3–5 times. Avoid plate drying between plate washings and prior to the addition of the next reagent. Tap each plate onto absorbent material after the final wash to remove any residual wash fluid;

5. BVDV, *M. paratuberculosis* and *E. coli* K99 ELISA plates with 100 μL of enzyme-labeled antibody per well, incubated at 18–26 °C for 30 min;

6. Repeat step 4;

7. Dispense 100 μL of TMB substrate solution into each well, incubated at 18–26 °C for 10 min (*M. paratuberculosis* 15 min);

8. Dispense 50 μL of stop solution into each well (BVDV and *M. paratuberculosis* is 100 μL);

9. Measure and record the absorbance of the samples and controls at 450 nm;

10. Calculation:

BVDV: Validity criteria: ODneg ≤0.25; ODpos-ODneg ≥0.15; S/P =(ODsample-ODneg)/(ODpos-ODneg); Interpretation: S/P < 0.2 is negative, 0.2 ≤S/P < 0.3 is suspect, S/P ≥0.3 is positive.

BRV and BCoV: Calculation of inhibition rate: %inh samples =[(ODneg-ODsamples)/ODneg]*100%, %inh positive =[(ODneg-ODpos)/ODneg]*100%; Validity criteria: ODneg-ODpo $s$ > 0.5, %inh positive >40%; Interpretation: %inh < 20% is negative, %inh ≥20% is positive.

*E. coli* K99: Calculation of inhibition rate: %inh samples =[(ODneg-ODsamples)/ODneg]*100%, %inh positive =[(ODneg-ODpos)/ODneg]*100%; Validity criteria: ODneg-ODpo $s$ > 0.7, %inh positive >50%; Interpretation: %inh < 20% is negative, %inh ≥20% is positive.

*M. paratuberculosis*: Validity criteria: ODpos ≥0.75, ODneg < 0.2; S/P =[(ODsamples-ODneg)/(ODpos-ODneg)]*100%; Interpretation: S/P ≤0.2 is negative, 0.2 < S/P < 0.3 is suspect, S/P ≥0.3 is positive.

## Data analysis

The map showing the location of the research herd was created using CAD software. According to VassarStats, the prevalence of diarrhea pathogens and the precise binomial 95% confidence intervals (Cis) were calculated respectively.

The GraphPad software was used for all statistical evaluations. $\chi^2$-tests and analysis of variance were used to determine differences in the prevalence rates between different regions and across age data. $p < 0.05$ was considered to represent a significant difference.

**Table 2  Samples information collected in this study.**

| Location | Leagues and cities | Number of scattered households | Total | Number of blood samples collected | | | |
|---|---|---|---|---|---|---|---|
| | | | | 0–2 months old | 2–6 months old | 6–18 months old | >18 months old |
| East | Chifeng | 4 | 145 | 50 | 30 | 25 | 40 |
| | Tongliao | 9 | 325 | 100 | 70 | 75 | 80 |
| | Hulun Buir | 7 | 233 | 90 | 35 | 48 | 60 |
| | Hinggan League | 8 | 355 | 105 | 80 | 80 | 90 |
| Central | Ulanqab | 7 | 329 | 90 | 70 | 69 | 100 |
| | Xilingol League | 3 | 146 | 55 | 30 | 31 | 30 |
| | Huhhot | 10 | 578 | 240 | 100 | 100 | 138 |
| West | Baotou | 6 | 275 | 90 | 60 | 60 | 65 |
| | Ordos | 5 | 200 | 65 | 45 | 35 | 55 |
| | Wuhai | 5 | 180 | 50 | 42 | 30 | 58 |
| | Bayan Nur | 8 | 284 | 95 | 63 | 42 | 84 |
| Total | 11 | 72 | 3,050 | 1,030 | 625 | 595 | 800 |

# RESULTS

## Herd research

According to the sampling strategy, 3,050 serum samples from 72 bovine herds in 11 leagues and cities in Inner Mongolia of China were collected for testing. We collected a minimum of five samples and a maximum of 60 samples from each individual household, 140–600 samples for each league city, with a total of 1,030 cattle aged 0–2 months, 625 cattle aged 2–6 months, 595 cattle aged 6–18 months and 800 adult cattle. Sample information collected is shown in Table 2.

## Detection of diarrhea-causing pathogen antibodies in blood samples

The infection rates for viruses and bacteria were 44% ($n = 1,342$) and 24.59% ($n = 750$), respectively. As shown in Table 3, Table S1, the serum total positive rate of at 0–2 months was 96.89%, which was significantly different from 2–6 months and 6–18 months ($p < 0.05$), and has no significantly different from adult cattle ($p > 0.05$). ETEC (enterotoxigenic *Escherichia coli*) adhesin gene K99 accounted for 13.80% of the total cases, and most cases (40.39%) occurred in cattle aged 0–2 months.

## Regional prevalence of bovine diarrhea pathogens antibody

Table 4 and Table S2 show that viral pathogens were detected in 11 leagues and cities of Inner Mongolia, while bacterial pathogens were detected in most of the leagues and cities. *E. coli* K99 was detected in 10 leagues and cities except Xilingol League, *M. paratuberculosis* was detected in 9 leagues and cities except Xilingol League and Wuhai. The positive rate of BVDV in Chifeng the highest at 67.59% ($p < 0.01$). The highest prevalence of BRV was in Xilingol League (48.63%) ($p < 0.01$). Xilingol League also had the highest prevalence rate of BCoV (51.37%) ($p < 0.01$). Baotou had the highest prevalence of *E. coli* K99 (45.09%) ($p$

**Table 3 Overall prevalence rates of the five pathogens.**

| Pathogen | Serum antibody positive rate (%) (95% CI) | Age | | | |
|---|---|---|---|---|---|
| | | 0–2 months (%) (95% CI) | 2–6 months (%) (95% CI) | 6–18 months (%) (95% CI) | >18 months (%) (95% CI) |
| BRV | 12.39 (11.27;13.61) | 20.58 (18.22;23.16) | 16.48 (13.78;19.59) | 8.57 (6.58;11.09) | 1.50 (0.86;2.6) |
| BCoV | 12.82 (11.68;14.05) | 22.91 (20.45;25.57) | 11.84 (9.54;14.61) | 4.87 (3.41;6.91) | 6.5 (4.99;8.42) |
| BVDV | 18.79 (17.44;20.22) | 13.01 (11.09;15.2) | 4.96 (3.52;6.95) | 2.02 (1.16;3.5) | 49.50 (46.05;52.96) |
| K99 | 13.80 (12.62;15.07) | 40.39 (37.43;43.42) | 0.80 (0.34;1.86) | 0 | 0 |
| *M. paratuberculosis* | 10.79 (9.74;11.94) | 0 | 2.56 (1.58;4.12) | 3.7 (2.46;5.54) | 36.38 (33.12;39.77) |
| Total | 68.59 (66.92;70.21) | 96.89[*] (95.64;97.79) | 36.64 (32.95;40.49) | 19.16 (16.2;22.52) | 93.88[ns] (92.00;95.34) |

$< 0.05$), and Hulun Buir had the highest prevalence rate of *M.paratuberculosis* (33.91%), which was significantly different from 9 leagues and cities except Hohhot ($p < 0.05$).

## Mixed infection

Table 5 shows that mixed infection of BVDV and other pathogens were present at all ages (957 positive samples, 31.38%), BRV+BCoV mixed infection was present at each age (9.74%). In young cattle (0–2 months old), the highest mixed infection rate was BVDV+BRV+*E. coli* K99 (13.40%). The highest mixed infection rate among 2–6-month-old and 6–18-month-old cattle was BRV+BCoV. The highest prevalence of of BVDV+BCoV+*M. paratuberculosis* infection (23.00%) was observed in adult cattle.

## DISCUSSION

The authors revealed that the prevalence of bovine diarrhea pathogens among the scattered households in Inner Mongolia of China. The detection of antibodies in serum samples is a fast and cost-effective method for detecting pathogens in infected herds (*Mõtus et al., 2021*). The detection kits used in this study are globally recognized products capable of detecting even low pathogen antibody levels (*Mõtus et al., 2021*). The authors found the infectious pathogens are complex, and that mixed infections comprising multiple pathogens are a serious issue, mainly affected cattle aged 0–2 months. BVDV is detected in 18.79% of cattle. There are several studies proving that BVDV is prevalent worldwide. In Tibet and Qinghai of western China, seropositivity rates were above 53% in yaks, and the BVDV positivity rate was 35–41% in Gansu Province, 77.8% in Eastern China, and 100% in Shanxi Province (*Gao et al., 2013*; *Hou et al., 2019*; *Qin et al., 2019*; *Ma et al., 2016*; *Deng et al., 2020*). *Ricci et al. (2019)* found that BVDV-1 was detected in unvaccinated cattle in

Xie et al. (2023), *PeerJ*, DOI 10.7717/peerj.16013

**Table 4** **The prevalence of the five pathogens among the different leagues and cities.**

| League and city | Chifeng city (%) (95% CI) | Tongliao city (%) (95% CI) | Hulun Buir city (%) (95% CI) | Hinggan League (%) (95% CI) | Ulanqab city (%) (95% CI) | Xilingol League (%) (95% CI) | Hohhot city (%) (95% CI) | Baotou city (%) (95% CI) | Ordos city (%) (95% CI) | Wuhai city (%) (95% CI) | Bayan Nur city (%) (95% CI) |
|---|---|---|---|---|---|---|---|---|---|---|---|
| BVDV | 67.59** (59.6;74.67) | 10.46 (7.58;14.26) | 26.18* (20.95;32.18) | 20.28* (16.43;24.77) | 9.73 (6.98;13.41) | 10.96 (6.86;17.06) | 25.43 (22.05;29.13) | 13.09 (9.61;17.59) | 10.00 (6.57;14.94) | 3.89 (1.90;7.81) | 17.61 (13.62;22.46) |
| BRV | 8.97 (5.32;14.74) | 7.08 (4.76;10.40) | 13.30 (9.53;18.26) | 8.17 (5.75;11.49) | 7.90 (5.45;11.32) | 48.63 (40.66;56.67) | 19.55 (16.52;22.98) | 4.00 (2.25;7.02) | 8.00 (4.98;12.60) | 9.44 (5.98;14.60) | 9.86 (6.91;13.88) |
| BCoV | 11.03 (6.90;17.17) | 7.69 (5.26;11.11) | 9.44 (6.32;13.88) | 7.32 (5.04;10.51) | 9.42 (6.72;13.06) | 51.37** (43.33;59.34) | 20.59 (17.49;24.08) | 6.55 (4.18;10.11) | 8.00 (4.98;12.60) | 10.00 (6.42;15.25) | 8.80 (6.03;12.67) |
| K99 | 8.97 (5.32;14.74) | 5.23 (3.29;8.22) | 19.74 (15.14;25.32) | 12.68 (9.61;16.55) | 5.78 (3.73;8.85) | 0 | 21.63 (18.47;25.17) | 45.09* (39.32;51.00) | 5.00 (2.74;8.96) | 1.67 (0.57;4.79) | 6.69 (4.32;10.21) |
| *M. paratuberculosis* | 4.83 (2.36;9.63) | 7.38 (5.01;10.75) | 33.91* (28.14;40.21) | 6.48 (4.36;9.54) | 5.78 (3.73;8.85) | 0 | 25.95 (22.54;29.67) | 3.64 (1.99;6.57) | 5.00 (2.74;8.96) | 0 | 2.46 (1.20;4.99) |

**Table 5  The prevalence of mixed infections among the bovines according to age.**

| Months | Pathogen | Number of serum samples | Number of positive serum for mixed infection | Positive rate (%) (95% CI) |
|---|---|---|---|---|
| 0–2 | BVDV+BRV | 1030 | 62 | 6.02% (4.72;7.64) |
| | BVDV+BCoV | 1030 | 106 | 10.29% (8.58;12.3) |
| | BRV+BCoV | 1030 | 87 | 8.45% (6.90;10.31) |
| | BVDV+BRV+BCoV | 1030 | 63 | 10.86% (8.50;13.75) |
| | BVDV+BRV+K99 | 1030 | 138 | 13.40% (11.46;15.62) |
| | BVDV+BRV+BCoV+K99 | 1030 | 108 | 10.49% (8.76;12.51) |
| 2–6 | BRV+BCoV | 625 | 82 | 13.12% (10.70;15.99) |
| | BVDV+BCoV | 625 | 42 | 6.72% (5.01;8.96) |
| | BVDV+BRV+BCoV | 625 | 32 | 5.12% (3.65;7.14) |
| 6–18 | BRV+BCoV | 595 | 52 | 8.74% (6.73;11.28) |
| | BVDV+BRV+BCoV | 595 | 27 | 4.54% (3.14;6.52) |
| | BVDV+BCoV+*M. paratuberculosis* | 595 | 10 | 1.68% (0.91;3.06) |
| >18 | BVDV+BCoV | 800 | 85 | 10.63% (8.68;12.96) |
| | BRV+BCoV | 800 | 76 | 9.50% (7.66;11.73) |
| | BVDV+*M. paratuberculosis* | 800 | 59 | 7.38% (5.76;9.40) |
| | BVDV+BCoV+*M. paratuberculosis* | 800 | 184 | 23.00% (20.22;26.04) |
| | BRV+BCoV+*M. paratuberculosis* | 800 | 140 | 17.50% (15.02;20.29) |
| | BVDV+BRV+BCoV+*M. paratuberculosis* | 800 | 40 | 5.00% (3.69;6.74) |

Italy (28%), and the positive rates was higher than 67% in Uruguay, Turkey and other countries (*Maya et al., 2020*; *Timurkan & Aydın, 2019*; *Aragaw et al., 2018*; *Nugroho et al., 2020*). Comparing the above percentages, the authors suggest that the positive rate may be due to the fact that none of the cattle collected were vaccinated against BVDV, the most BVDV positive sera were detected in adult cattle (49.50%), probably due to natural infection or immunity produced by the organism itself after resistance.

In this study, BVDV antibodies were detected, which may indicate that the virus is widespread in herds across different ages. Scattered households practice mixed rearing, resulted in the active cross-infection of the BVDV pathogen in cattle herd. In cattle farming, BVDV vaccination should be used to protect susceptible cattle (*Marschik et al., 2018*; *Pinior et al., 2017*; *Richter et al., 2017*; *Richter et al., 2019*).

Interestingly, the authors found that the antibody positive rate was generally higher in cattle aged 0–2 months ($p < 0.01$). *E. coli* K99 mostly occurs in calves that are several days old. The ability of the *E. coli* K99 antigen to bind to the mucosa of the small intestine

depends on the age of the animal and gradually decreases from 12 h after birth (*Runnels, Moon & Schneider, 1980*). The *E. coli* K99 positive rate among bovines aged 0–2 months 40.39%, may be related to the immune gap caused by the reduction of maternal antibodies, and that the antibody protection of calves is not sufficient (*Acres, 1985*; *Al Mawly et al., 2015*). The decrease in maternal antibodies can also explain the increase in the prevalence of BCoV at this stage (*Tzipori, 1985*). BRV is also mainly concentrated among bovine aged 0–2 months, which is due to the short incubation period of BRV, and the high susceptibility of bovine within 0–2 months of age (*Snodgrass et al., 1986*; *Uhde et al., 2008*).

The authors revealed an age dependence of the prevalence of these pathogens based on the consistent age-related trends of individual pathogens and mixed infections detected. The mixed infection of BVDV+BRV+*E. coli* K99 in 0–2 months and BVDV+BCoV+*M. paratuberculosis* in adult cattle follows the epidemic process related to age. The test results also indicate a long-term sensitivity and synergy between BRV and BCoV. This finding can be explained by cytoskeleton changes caused by virus infection, which is related to the reduction of disaccharidase top expression (*Brunet et al., 2000b*; *García et al., 2000*), previous studies have shown that the decrease of disaccharidase activity in cells may promote the growth of bacteria whether it causes cell damage or not (*Torres-Medina, Schlafer & Mebus, 1985*; *García et al., 2000*; *Reynolds et al., 1986*), which may explain the high frequency of mixed infections with viral and bacterial pathogens in Inner Mongolia. In the case of a BRV endotoxin infection, NSP4 is produced in the cell, and the up-regulation of $Ca_2^+$ affects the $Ca_2$ sensitive protein F-actin, villin and tubulin, resulting in damage to the microvilli cytoskeleton of the cell (*Brunet et al., 2000a*; *Brunet et al., 2000b*; *Collins et al., 1988*; *Durham, Farquharson & Stevenson, 1979*) which leads to the infection of other pathogens. The diversification of mixed infections indicates that the needs of scattered households are often different and more diversified (*Brunauer, Roch & Conrady, 2021*; *Dahmani et al., 2020*; *Mõtus et al., 2021*).

There are great differences in the pathogen types and prevalence of infection in different leagues and cities, and the main pathogens of their epidemics are also different indicating that these pathogens causing bovine diarrhea have significant regional differences. In most leagues, the five pathogens have been detected, indicating that these are widely prevalent in free range households in Inner Mongolia. Chifeng, Tongliao and Wuhai are mostly beef cattle, the feeding method is intensive management and the slaughter time is fast. Generally, cattle were slaughtered in approximately 18 months to 24 months, which has more opportunities for cattle transportation. Hohhot, Hinggan League, Baotou, Bayan Nur, and Ordos are dominated by dairy cattle, most of which are self-raising and self-reproductive, with high breeding density. Ulanqab, Hulun Buir, and Xilingol leagues mainly relying on natural pastures, with no thermal insulation measures in place, which is also the reason for the differences in the geographical prevalence of the pathogens among bovine diarrhea. In the process of cattle feeding, the importance of good feed and pasture, as well as the importance of preventive treatment, such as accurate colostrum intake, protecting the herd from the cold and heat, keeping the cattle farm clean, vaccinating on time, and disinfecting the cattle before returning from grazing. Responsible herders and good grazing methods are also considered effective preventative measures. Moreover, in the

process of pasture grazing, it is necessary to maintain a large amount of high-quality grass, shade, water and maintain health, if there is insufficient water or grass, supplementary feed is required (*Chenais et al., 2021*; *EFSA Panel on Animal Health and Welfare, 2012*).

This is the first study that detected and analyzed the main pathogen antibodies that cause bovine diarrhea in 72 free-range households in 11 leagues and cities of Inner Mongolia. The focus of this study was on bovine of different months of age in scattered households in the Inner Mongolia of China, and the interaction of several pathogens was considered. The results showed that the prevalence of mixed infections and their important influencing factors are quite different. The authors clarified the main pathogens and epidemiological trends of bovine diarrhea in Inner Mongolia of China, which may help expand understanding of the epidemiological situation of bovine diarrhea and formulate effective disease prevention and control strategies, it is important for improving the control of disease occurrence and achieving the goal of improving animal health.

## CONCLUSIONS

The current research shows that the cattle herds of scattered households in Inner Mongolia are endemic infected with BVDV, BRV, BCoV, *E. coli* K99, and *M. paratuberculosis*. The results showed that the highest positive rate among the pathogens was BVDV at 18.79%, the highest prevalence of BVDV, BRV, BCoV, and *E. coli* K99 was found in cattle aged 0–2 months, the highest prevalence of *M.paratuberculosis* occurred in adult cattle, and the difference in prevalence of the five pathogens varied significantly across the 11 leagues and cities. The most common mixed infection was BRV+BCoV. The disease-specific patterns revealed in this study provide valuable data and information for in-depth understanding of pathogen behaviors in a wide range of cattle, and can be used as a reference for formulating disease control plans.

## ACKNOWLEDGEMENTS

We are grateful to the Inner Mongolia Animal Disease Control Center staff for assisting in the collection of some samples and providing background information on those samples.

### Funding
This work was supported by the Inner Mongolia Autonomous Region Major Science and Technology Project (2020ZD0006, 2021ZD0013) and the Major Science and Technology Project of Hohhot (2022-Agriculture-Major-1-1). The funders had no role in study design, data collection and analysis, decision to publish, or preparation of the manuscript.

### Grant Disclosures
The following grant information was disclosed by the authors:
Inner Mongolia Autonomous Region Major Science and Technology Project: 2020ZD0006, 2021ZD0013.
Major Science and Technology Project of Hohhot:  2022-Agriculture-Major-1-1.

## Competing Interests

The authors declare there are no competing interests.

## Author Contributions

- Mengyuan Xie conceived and designed the experiments, performed the experiments, analyzed the data, prepared figures and/or tables, authored or reviewed drafts of the article, and approved the final draft.
- Kejia Chen performed the experiments, prepared figures and/or tables, samples collected, and approved the final draft.
- PingPing Liu performed the experiments, prepared figures and/or tables, samples collected, and approved the final draft.
- Xiaodan Wang performed the experiments, prepared figures and/or tables, samples collected, and approved the final draft.
- Yexin Chen performed the experiments, prepared figures and/or tables, samples collected, and approved the final draft.
- Hewei Shang performed the experiments, authored or reviewed drafts of the article, samples collected, and approved the final draft.
- Yanru Hao performed the experiments, prepared figures and/or tables, samples collected, and approved the final draft.
- Peiyuan Gao performed the experiments, prepared figures and/or tables, samples collected, and approved the final draft.
- Xiuling He analyzed the data, authored or reviewed drafts of the article, and approved the final draft.
- Xiaojing Xu conceived and designed the experiments, analyzed the data, authored or reviewed drafts of the article, and approved the final draft.

## Data Availability

The data is available in the Supplemental Files.

## Supplemental Information

Supplemental information for this article can be found online at http://dx.doi.org/10.7717/peerj.16013#supplemental-information.

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
