# Peer review of "Seroprevalence of five diarrhea-related pathogens in bovine herds of scattered households in Inner Mongolia, China between 2019 and 2022"

_PeerJ, doi:10.7717/peerj.16013_

## Round 0.1 · original submission · Major Revisions

Dear Dr. Xie and colleagues:

Thanks for submitting your manuscript to PeerJ. I have now received three independent reviews of your work, and as you will see, the reviewers raised some concerns about the research. Despite this, these reviewers are quite optimistic about your work and the potential impact it will have on research studying ruminant pathogens associated with households. Thus, I encourage you to revise your manuscript, accordingly, taking into account all of the concerns raised by all three reviewers.

While the concerns of the reviewers are relatively minor, this is a major revision to ensure that the original reviewers have a chance to evaluate your responses to their concerns. There are many suggestions, which I am sure will greatly improve your manuscript once addressed.

There are many minor suggestions to improve the manuscript (typos, nuances, etc.). Reviewer 2 has kindly provided edits on your manuscript.

Therefore, I am recommending that you revise your manuscript, accordingly, taking into account all of the issues raised by the reviewers.

I look forward to seeing your revision, and thanks again for submitting your work to PeerJ.

Good luck with your revision,

-joe

Reviewer 1 ·

Basic reporting

The manuscript (MS) entitled “Cross-sectional study on the pathogens responsible for diarrhea in the bovine herds of scattered households in Inner Mongolia” reports the seroprevalence of five bovine diarrhea-related pathogens in bovine herds of scattered households in Inner Mongolia, China. The manuscript was relatively well designed and executed, and deserves publication after major revision. The title suggest as “Seroprevalence of five diarrhea-related pathogens in bovine herds of scattered households in Inner Mongolia, China between 2019 and 2022”.

Experimental design

Only seroprevalence of five diarrhea-related pathogens in cattle were detected with ELISA in this study, the five diarrhea-related pathogens were not detected.

Validity of the findings

The results of this study were only related with the seprevalence, not directly with pathogens. Meanwhile, the manufacturers were not provided for the commercial ELISA kits used in this study. The detection results of fecal samples were also not provided.The description of fecal samples suggest to be deleted. The “Discussion” should be shortened.

Additional comments

Specific comments
Throughout the MS, there are some errors. The following suggestions should be noted.
1.Lines 66-99 should be shortened.
2. Line 91-95 “that end, this study assessed the prevalence of five infectious pathogens in blood samples in Inner Mongolia, China, and determined the epidemiological and epidemiological differences of
the pathogens causing bovine diarrhea among 11 leagues and cities in Chinaís Inner Mongolia so as to provide guiding suggestions for the prevention and control of Chinese bovine diarrhea diseases” suggest as “that end, this study assessed the seroprevalence of five diarrhea-realted pathogens in cattle in Inner Mongolia, China for providing guiding suggestions for the prevention and control of bovine diarrhea”.
3. Lines 130,131 “vaccination, clinical symptoms, and other information, for each pathogenic infection, through the analysis of the collected serum samples were collected” suggest as “vaccination, clinical symptoms, and each pathogenic infection were collected.”
4. Line 137 “The sterile cotton swab were used to rotata into the anus to collect fecla samples and put” suggest as “The sterile cotton swabs were used to rotate into the anus to collect fecal samples and put”.
5. Line 213 “L. Maya et al. collected 2,546 serum” suggest as “Maya et al. collected 2,546 sera”.

Reviewer 2 ·

Basic reporting

The manuscript needs an extensive revision for all parts

Experimental design

The research question is not well defined and the methodology needs to be revised (see the attached annotated file"

Validity of the findings

The manuscript needs to be revised and the conclusion should be revised accordingly

Additional comments

All my coments are in the annotated attached file (the authours should tak in consideration al my comments of this file)

Annotated reviews are not available for download in order to protect the identity of reviewers who chose to remain anonymous.

Reviewer 3 ·

Basic reporting

In the manuscript 78043, authors performed a cross-sectional study on pathogens responsible for diarrhea in the bovine herds of scattered households in Inner Mongolia. The findings are of practical significance for prevention and control of bovine diarrhea in the Inner Mongolia, China. Overall, the manuscript was well written and organized. The results were well presented. However, some issues especially in validity of findings required to be clarified. Based on these, I suggest to give a major revision.

Experimental design

Well done.

Validity of the findings

1. T test is usually used to determine statistic differences between two groups. It is not proper to use T test to determine differences in the prevalence rates between different regions and across age.
2. Figures of high quality should be provided in the manuscript.
3. It seemed that authors just used ELISA kits to detect specific pathogen antibody in serum samples to define positive or negative. Did authors use molecular based diagnosis such as PCR or qPCR to identify the results? Authors have collected fecla samples and extracted the genome using DNA/RNA extraction kits. I think that it would be more rigorous to combine the serological testing and molecular testing results to determine positive or negative samples. So, authors may need to re-analyze the results if possible.

Additional comments

Lines 140-141, Table 1 is not the ELISA kits information. Authors need to provide ELISA kits details in the manuscript.

---

## Round 0.2 · Minor Revisions

Dear Dr. Xie and colleagues:

Thanks for revising your manuscript. The reviewers are generally satisfied with your revision (as am I). Great! However, per reviewer 2 there are a few concerns to address. Please attend to these issues ASAP so we may move towards acceptance of your work.

Best,

-joe

Reviewer 1 ·

Basic reporting

no comment

Experimental design

no comment

Validity of the findings

no comment

Additional comments

No comment

Reviewer 2 ·

Basic reporting

The manuscript needs an extensive revision

Experimental design

The methods should be clearly defined

Validity of the findings

the analysis should be revised

Additional comments

I would like to thank the authors for their efforts to improve the quality of the manuscript. However, The manuscript needs more and more efforts to be acceptable for publication:
My main concern is related to the fact that the authors did not precise if they sampled diarrheic or non diarrheic animals. Please precise
The second concern is related to the discussion which is practically unreadable and the authors have not made any efforts to revise or summarize it. The discussion is mainly written as: this autogiros found, …., the authors revealed… (try to summarize please).
I have also other comments:
Abstract:
Line 23: "….the prevalence of bovine diarrhea pathogens…" (delete found please)
Line 28: BVDV, BRV, BCoV, K99: define please
K99 should be E coli K99 (K is just and antigen): revise in all the manuscript please
Line 36: At 0-2 months : correct please (expl: in animal aged, in the category of…..)
Line 36-38: The prevalence of BVDV and M. paratuberculosis was the highest in adult cattle which was significantly different from that in other age groups (p <0.01). revise please (exp: was significantly higher in adults than in the other categories…(you can use any other correct sentence)).
Line 38: the five disease may be the five "pathogens"
Line 39: age stage (replace by age category)
Lines 41 and 43: Write the cattle as : cattle (without the)
Line 44: Economic losses (not loss).
Introduction:
Line 55: and Bovine viral…(add "and")
Line 56: add"aznd before Salmonella
Line 59: symptom should be symptoms
Line 66: the added reference is not in relation with your explanation (and what you explain in the rebuttal letter, Dahmani et al is from Algeria and you report data from China. Delete it please
Lines 72-79: add a reference please
Lines 87-94: the objectives should be revised to delete the repetition
Materials & Methods
Line 113-114: "blood…." delete please (the sampling method is described in line 118)
Line 119: vaccination : write as vaccination status
Line 130: you should describe the realization of the technique. How did you proceed? How can one understand what did you do exactly?
Line 135: the t test is not adapted to compare proportion
Additionally, we did not find the results of your statistical analysis in any table
Results
Line 147: bacterium should be bacteria
Line 150: ETEC: define it
Line 153: "table 3, ……shows…." . avoid to use and repeat the same expression
Line 153:…in 11 cities…" you can say that they were detected in all cities…
Line 152-159: you should confirm your results with the statistical analysis (and in all your results).
Lines 162-167: try to reformulate your sentences please (exp: line 164: each ages (may be each age); Line 166-167…).
Discussion:
Your discussion is practically unreadable and needs an extensive revisions (as explained before).
Line 170: bovine diarrhea pathogens (not disease)
Line 173: how did bovine diarrhea impact future epidemics?
Line 174-176: add a reference
Line 179: this study revealed five major pathogens (they are the only pathogens searched in your study)
Line 181: of bovine?
Line 184: abortion disease feeds?
Line 194: found that positive rate of ?
Line 196:Ma et a.
Line 199: BTM
Line 218: decreased significantly (where are the results of the statistical analysis?)
These are just some remarks
Briefly, the discussion need to be COMPLETELY revised and SUMMARIZED
Revise the last parts of your manuscript
Acknowledgements, authors contributions….they are unreadable
Revise your references
The name of bacterial species should be in Italic (E coli, C parvum..)
Complete reference 4 (Brunauer….)
Chenais, E., Wennström, P., Kartskhia…..: the PMID number is not correct
Figure 1:
Improve the quality of the figure and add the name of the cities/leagues.
Table 2 :
Improve the quality of the tables (all your tables)
Make "old" in the same line
Add the percentage between brackets (%) for all your numbers. It will be more interesting
Add a line to separate between the different regions.
Add the results of the statistical analysis
Figure 2:
Since you have chosen to add the other tables delete this figure. We cannot put a table and a figure for the same result. Additionally, the figures are not very clear
Tables 2,3 and 4:
Separate between the categories and add the results of the statistical analyses

---

## Round 0.3 · Minor Revisions

Dear Dr. Xie and colleagues:

Thanks for again revising your manuscript. Reviewer 2 is generally satisfied with your revision (as am I). Great! However, there are a few concerns to address (see edited manuscript). Please attend to these issues ASAP so we may move towards acceptance of your work.

Best,

-joe

Reviewer 2 ·

Basic reporting

See the attached file

Experimental design

/

Validity of the findings

/

Additional comments

See the attached file

Annotated reviews are not available for download in order to protect the identity of reviewers who chose to remain anonymous.

---

## Round 0.4 · accepted · Accept

Dear Dr. Xie and colleagues:

Thanks for revising your manuscript based on the concerns raised by the reviewers. I now believe that your manuscript is suitable for publication. Congratulations! I look forward to seeing this work in print, and I anticipate it being an important resource for groups studying ruminant pathogens associated with households. Thanks again for choosing PeerJ to publish such important work.

Best,

-joe